# VeriLoC: Line-of-Code Level Prediction of Hardware Design Quality from Verilog Code

**Raghu Vamshi Hemadri**[1*]   **Jitendra Bhandari**[1*]   **Andre Nakkab**[1]   **Johann Knechtel**[2]
**Badri P Gopalan**[3]   **Ramesh Narayanaswamy**[3]   **Ramesh Karri**[1]   **Siddharth Garg**[1]

[1]New York University Tandon School of Engineering
[2]New York University Abu Dhabi
[3]Synopsys

## Abstract

Modern chip design is complex, and there is a crucial need for early-stage prediction of key design-quality metrics like timing and routing congestion directly from Verilog code (a commonly used programming language for hardware design). It is especially important yet complex to predict individual lines of code that cause timing violations or downstream routing congestion. Prior works have tried approaches like converting Verilog into an intermediate graph representation and using LLM embeddings alongside other features to predict module-level quality, but did not consider line-level quality prediction. We propose `VeriLoC`, the first method that predicts design quality directly from Verilog at both the line- and module-level. To this end, `VeriLoC` leverages recent Verilog code-generation LLMs to extract local line-level and module-level embeddings, and trains downstream classifiers/regressors on concatenations of these embeddings. `VeriLoC` achieves high F1-scores of 0.86–0.95 for line-level congestion and timing prediction, and reduces the mean average percentage error from $14\% - 18\%$ for SOTA methods down to only $4\%$. We believe that `VeriLoC` embeddings and insights from our work will also be of value for other predictive and optimization tasks for complex hardware design.

## 1   Introduction

Modern chip design is highly *complex*. It begins with devising a description of the chip's behavior in a hardware description language (HDL) like Verilog.[2] This is followed by a series of automated steps, including synthesis (where RTL code is converted into a circuit of Boolean logic and its gate implementation), placement (which arranges gates on the chip canvas), and routing (which connects gates using metal wires). This process transforms the RTL code into a manufacturable chip layout.

Key metrics for design quality, like area, timing, power, routing congestion, *etc.*, can only be verified from final layouts, but obtaining these layouts can take hours or days as synthesis, placement, routing, and other steps in the design flow are extremely complex and time-consuming. Designers often iterate multiple times till specifications and quality targets are met; these iterations can take anywhere from weeks to months, impacting time-to-market. Timing and routing congestion, in particular, are difficult to manage and are frequently the main impediment to design closure  [1–5].

---

[*]Both authors contributed equally to this research.

[2]HDL codes are commonly also referred to as register-transfer level (RTL) descriptions. We use RTL or Verilog interchangeably from here on.

39th Conference on Neural Information Processing Systems (NeurIPS 2025).

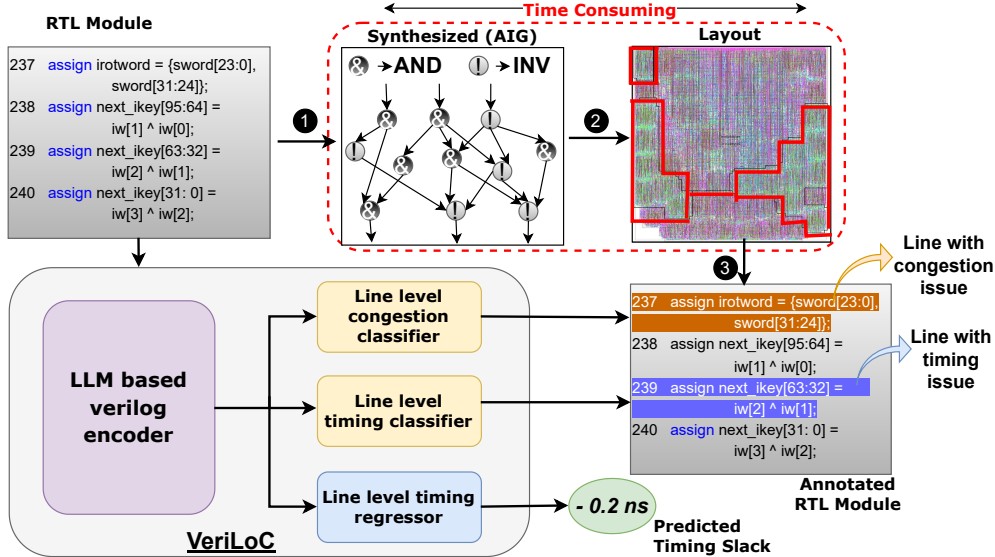

Figure 1: Conventional flow vs proposed for an exemplary AES design. ❶ The RTL code is converted into a synthesized netlist, e.g., represented by an AIG. ❷ The netlist is converted into a layout, with congestion arising in green areas (bounded in red). ❸ Congestion information is annotated and traced back to the RTL. With VeriLoC, we directly predict congestion and timing for the RTL at run-time, bypassing the time-consuming conventional steps.

To address this issue, a body of recent work has proposed *early-stage prediction* of design quality from RTL code, typically via intermediate representations of RTL like and-inverter graphs (AIGs)[3] [6–10]. However, intermediate representations might lose rich semantic information available in the RTL code in compact form. For example, a 64-bit multiplier is a single line of Verilog, but corresponds to hundreds or thousands of gates in an AIG, where information must then be *reverse engineered* by the ML model. Recent work has leveraged large language model (LLM) based encodings of Verilog modules for accurate module-level power, performance, and area (PPA) prediction [11, 12].

Aside from early-stage module-level predictions, designers can greatly from identifying *individual lines of code* (LoC) responsible for inducing timing violations or routing congestion. While electronic design automation (EDA) tools like RTL Architect [13] provide the capability of back-annotating the lines of code from a final layout (Step 3 in Fig. 1), these too are complex and time-consuming. Here, we pose a new research question that has not been addressed in literature: *can we predict design quality, specifically, timing and routing congestion, from RTL code at the module and the individual line-of-code level?*

A key challenge in addressing this question is how to obtain informative RTL embeddings—here, we leverage the recent emergence of LLMs trained specifically for Verilog code generation like *CL-Verilog* 13B [14]. Although *CL-Verilog* is a decoder-only model, recent studies [15, 16] have demonstrated that internal activations from these models can yield effective embeddings. Using penultimate layer outputs from *CL-Verilog* as embeddings, we propose VeriLoC, a novel architecture for line-level classification of Verilog code. To the best of our knowledge, the LoC-level prediction problem has not been addressed in literature before.

The key idea in the proposed VeriLoC architecture (Figure 2) is to concatenate embeddings of each line-of-code in a Verilog module with an embedding of the entire module, thus obtaining both a local and global context. In practice, we find that additionally concatenating embeddings from up to two *neighboring* lines further improves performance. VeriLoC then trains a supervised classifier (or regressor) on ground-truth data from Synopsys RTL Architect [17] on the OpenABCD dataset [18],

---

[3]An AIG is a Boolean circuit, which consists only of the so-called universal set of AND and NOT gates, and is at the same time functionally identical to the RTL.

using models like XGBoost [19] and LightGBM [20] tailored for scarce and imbalanced data. Our contributions are as follows:

- We propose and evaluate `VeriLoC`, a novel LLM-based architecture for early-stage prediction of hardware design quality *directly* from RTL code, both at the individual lines of code and entire modules. Prior work only performs module-level predictions and converts RTL to an intermediate representation, thereby losing rich semantic information.
- We identify the importance of capturing both the local context, i.e., neighboring lines of code, and global context, i.e., the entire Verilog module, in enabling line-level timing and congestion predictions. `VeriLoC`'s architecture concatenates both local and global embeddings before the final classification step.
- `VeriLoC` achieves F1-scores of 0.86 in line-level congestion prediction, 0.95 in line-level timing prediction, and also outperforms state-of-art in module-level timing prediction, reducing the mean average percentage error from $18\%$ and $14\%$ to only $4\%$.
- We demonstrate the usefulness of LLMs specialized for RTL code generation to also generate powerful RTL code *embeddings* that can be used for challenging downstream prediction tasks, specifically, timing and routing congestion prediction.

Overall, `VeriLoC`[4] establishes an entirely new approach for early-stage prediction from RTL code, which might be of value not only to other prediction tasks, but also for code and design optimization.

## 2 Background and Related Work

We discuss relevant background on hardware design and contrast `VeriLoC` with related work on predicting design quality and on LLM-based prediction of code quality.

### 2.1 Hardware Design: Quality Metrics and Prediction

#### 2.1.1 Routing Congestion

**What is Routing Congestion?** Routing is one of the most complex and time-consuming steps in hardware design. Routing entails interconnecting logic gates with wires after the gates are placed on the chip canvas. Modern chips have tens of different routing layers, where any two wires that need to cross without connecting electrically can be routed above another, akin to a flyover in a traffic network. Almost every problem related to routing is known to be intractable [21]. Thus, like most processes in EDA, routing is heavily reliant on heuristic optimization, which cannot guarantee best quality in one go. In this context, managing *routing congestion*—or congestion for short—is important. This arises when multiple wires pass through the same small area of a chip, such that, in the worst case, the number of routing layers is insufficient to route all wires correctly, i.e., without at least two wires crossing paths. When this happens, the entire design might need to be undergo placement again, or might even necessitate an RTL rewrite.

**Predicting Routing Congestion.** Traditional methods commonly integrate actual routing processes [22–25] or analytical models that estimate congestion [26–29]. However, routing-based methods are plagued by considerable runtime cost while analytical-based approaches suffer from relatively low accuracy. To address these challenges, more recent works have employed ML techniques. For example, [30] utilize convolutional neural networks (CNNs) to predict the overall routability of placement solutions. In a follow-up work, [31] employ deep neural networks (DNNs) to achieve better performance for congestion prediction and guide toward less-congested placement. Furthermore, [32–37] all use graph neural networks (GNNs) using synthesized and/or placed netlists as inputs, which require running time-consuming synthesis and placement tools, respectively.

#### 2.1.2 Timing

**What is Timing?** Timing is a critical aspect of chip design and determines the fastest frequency at which the chip can operate. Timing is affected by every process in the design cycle, but the most critical impact is within the RTL stage, as this dictates the architecture and data flow of the IC. Roughly, timing refers to the time it takes for data to propagate from a circuit's input to its output;

---

[4]https://github.com/ML4EDA/VeriLoC.git

thus, the longest sequence of gates from the input to the output is referred to as the critical path. Often, timing is measured by *worst negative slack (WNS)*, i.e., the difference/slack between the desired critical-path delay and the actual critical-path delay. The goal for designers is to push WNS above zero, i.e., to keep delays within the desired budget.

**Timing Prediction.** Prior works predict timing by employing various ML techniques at various design stages. Closest to our work, [1–4] predict timing for entire modules at the RTL stage, but use either AIGs or other intermediate representations. Of these, [4, 1] are the current state-of-the-art methods. `VeriLoC` demonstrates substantial accuracy improvements compared to both. Other methods propose timing prediction at later stages in the design, including after synthesis [5] and after placement [38, 39]. All of these methods are focused on module-level timing prediction; `VeriLoC` is the first method to provide line-of-code level predictions of WNS.

## 2.2 LLMs for Code Generation and Quality Prediction

### 2.2.1 LLMs for Software Code Quality

LLMs have demonstrated significant potential for coding, with applications spanning bug detection, program synthesis, and performance optimization [40]. Models like CodeBERT, GraphCodeBERT, and CodeT5 effectively capture syntactic and semantic nuances in high-level programming languages, making them invaluable for tasks like code summarization, translation, and repair [41, 42]. The aforementioned LLMs excel at these generative tasks. Bug detection can be viewed as a line-level prediction task, and has been addressed via pattern matching using static analysis [43], enhanced with LLMs [44], or by using LLMs for test generation and fuzzing [45]. In most instances, given the massive amounts of open-source software and vulnerability datasets, these methods can leverage LLMs with careful prompt tuning, retrieval, and agentic frameworks. Indeed, state-of-art approaches like LLMSAN [46] utilize few-shot chain-of-thought prompting to extract structured data-flow paths for bug detection, but do not make any architectural modifications. Unfortunately, data is scarce in hardware, and concepts like routing congestion are barely mentioned. As we show later, prompting methods fail completely for line-level congestion and timing estimation.

### 2.2.2 LLMs for Hardware

While LLMs originally targeted software code, recent work has extended their use to hardware description languages such as Verilog. Generative Verilog models (e.g., VeriGen [47], CLVerilog [14], RTLCoder [48] and Others [49–52]) achieve impressive synthesis quality but do not provide downstream quality-of-results (QoR) metrics. Building on this trend, RTLRewriter applies LLM-guided rewriting for optimization [53], and RTLFixer employs LLM-driven debugging to correct syntax errors at scale [54]. Beyond generative tasks, LLMs have begun to assist QoR estimation for rapid design-space exploration. For PPA estimation, multimodal techniques—including CircuitFusion and VeriDistill hardware code with structural or graph-based embeddings to predict power, performance, and area [12, 11]. However, these methods operate at module or graph granularity, leaving line-level semantics unexplored. To the best of our knowledge, `VeriLoC` is the first ever line-level QoR predictor using a hardware-specialized LLM, enabling prediction of timing and congestion metrics directly at the statement level in Verilog code.

## 3 Methodology

**Overview.** `VeriLoC` builds on the premise that LLMs customized for RTL/Verilog code generation that have recently begun to emerge can be also be used as embeddings that capture the semantics of RTL code and used for downstream prediction tasks. `VeriLoC` is the first to demonstrate this property in the hardware context. We illustrate our methodology in Fig. 2. Our approach relies on embeddings generated by *CL-Verilog* [14], a variant of LLaMA-2 fine-tuned on Verilog code, extracted from its penultimate layer activations. We use these embeddings hierarchically, offering semantic representations at both the **module-** (Sec. 3.1) and **line-level** (Sec. 3.2), potentially with more context from neighboring lines (Sec. 3.2). These embeddings are projected to a lower dimension (Sec. 3.3), concatenated and a final classification/regression head outputs line- and module-level predictions (Sec. 3.5).

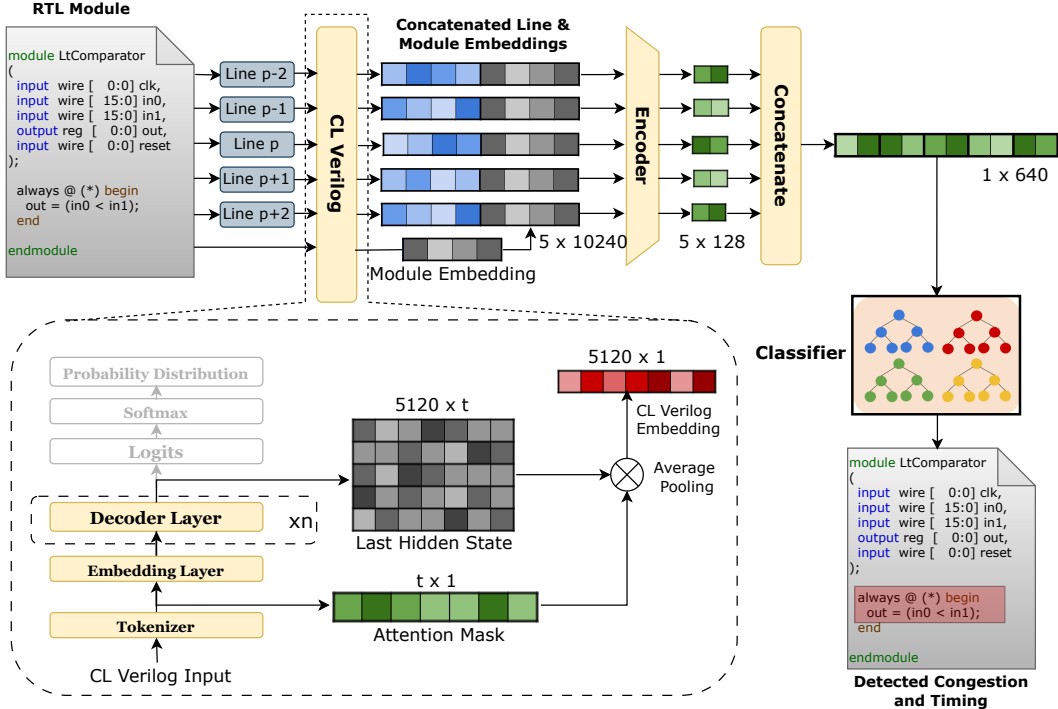

Figure 2: The architecture of `VeriLoC` for line-level timing and congestion prediction from RTL. Module-level prediction uses module embedding. The context window is set to $p = 5$ in this example.

## 3.1 Module-Level Embeddings

Modules in RTL designs are represented as sequences of lines of Verilog: $M = \{l_1, l_2, \ldots, l_n\}$, where $l_i$ is the $i$-th line of code in the module. To capture the global semantics of the module, the complete module is passed through *CL-Verilog*, which generates hidden states $H = \{h_1, h_2, \ldots, h_n\}$, where $h_i \in \mathbb{R}^k$ is the latent vector for the $i$-th token of *CL-Verilog*'s tokenizer, and $k$ is the dimensionality of the model's hidden state. Module embeddings $e(M)$ are computed as the pooled dot products of the hidden states and attention mask, normalized by sum of the attention mask:

$$e(M) = \frac{\sum_{i=1}^{n}(h_i \cdot m_i)}{\sum_{i=1}^{n} m_i}.$$

where $m_i \in \{0, 1\}$ is the attention mask that ensures only valid tokens contribute to the embedding, and the dot product $h_i \cdot m_i$ highlights the importance of each hidden state relative to the mask. The resulting module embedding $e(M) \in \mathbb{R}^k$ provides a condensed global representation of the module, enabling the detection of macro-level patterns such as resource utilization and timing violations.

## 3.2 Line-Level Embeddings

To capture localized semantics of Verilog and their impact on design quality, embeddings are also generated for individual lines of code. Each line $l_i$ is passed independently through *CL-Verilog*, producing a hidden state $h_i$. Similar to module embeddings, the line embedding $e(l_i)$ is computed using the attention-weighted pooling mechanism:

$$e(l_i) = \frac{\sum_{i=1}^{n}(h_i \cdot m_i)}{\sum_{i=1}^{n} m_i}.$$

These embeddings $e(l_i) \in \mathbb{R}^k$ focus on the specific performance characteristics of each line, such as whether it contributes to congestion or WNS.

### 3.3 Dimensionality Reduction

After extracting module-level and line-level embeddings, dimensionality reduction is applied to ensure computational efficiency and improve downstream task performance. The combined embedding $x_i = [e(l_i); e(M)]$ undergoes dimensionality reduction as follows.[5] An encoder-decoder framework is trained to reconstruct the original concatenated embedding $x_i$ from its reduced representation. The encoder maps the high-dimensional input $x_i$ to a lower-dimensional space $\mathbb{R}^d$: $z_i = Encoder(x_i)$, while the decoder reconstructs $x_i$ from $z_i$: $\Phi(x_i) = Decoder(Encoder(x_i))$. The framework is optimized to minimize the reconstruction loss: $\mathcal{L} = \|x_i - \Phi(x_i)\|^2$. Once training is complete, only the encoder is retained for dimensionality reduction. In short, the encoder provides compact embeddings $z_i$ and serves as a pre-trained initialization for downstream classification and regression.

### 3.4 Contextual Feature Augmentation

Contextual feature augmentation enhances the representation of a target line by integrating dependencies from surrounding lines. It captures sequential patterns and inter-line relationships, which are essential for analyzing RTL code. The embeddings of a target line $l_i$ are concatenated with those of its neighboring lines within a context window $p$. For any $l_i$, the augmented embedding is: $z_{\text{aug}}(l_i) = [z_{i-p}; \dots; z_i; \dots; z_{i+p}]$, where $z_i$ is the reduced embedding and $[\cdot]$ denotes vector concatenation. This approach introduces local dependencies, enabling the classifier to capture the sequential nature of RTL designs.

As shown in Fig. 3 (left), the statement `always @(posedge clk) begin` is not flagged, but in Fig. 3 (right), with `maybe_full <= N17;` introduced, the same line `always @(posedge clk) begin` becomes congestion-causing. This shows context-aware analysis can enhance detection by considering dependencies between neighbouring lines.

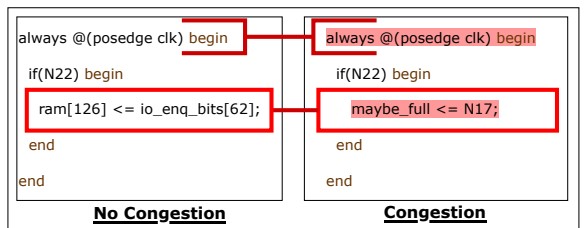

Figure 3: Effect of neighbor embeddings in context-aware congestion and timing detection.

### 3.5 Classification and Regression Heads

The concatenated embeddings feed into a classification head for line-level classification of code that cause congestion and timing issues, and a regression head for WNS prediction. We compare three classification/regression heads: (1) **Feedforward Neural Networks (FNNs)**, a single-layer and fully connected neural network that replaces *CL-Verilog*'s original classification head, but with a single classification (or regression) output; (2) **XGBoost**, a gradient-boosted tree [19]; (3) **LightGBM**, a lightweight gradient-boosting framework optimized for speed and performance [55]. XGBoost and LightGBM are used because of their demonstrated performance on imbalanced datasets [56]. This is essential for our work as only a small number of lines of code cause congestion or timing issues.

Although our primary focus is LoC-level prediction, we also use `VeriLoC` to estimate *module-level* WNS. Specifically, we we first estimate WNS at the line level and then select the worst (smallest) value across all lines. This line-wise granular prediction strategy allows the model to capture granular WNS estimates, and improves upon state-of-art module-level WNS predictors that use only module-level embeddings or features. This formulation, to the best of our knowledge, is unique to `VeriLoC`.

## 4 Empirical Evaluation

### 4.1 Experimental Setting

**Dataset.** We use the popular OpenABCD [18] RTL/Verilog code dataset for our experiments, using various Verilog modules from all projects in the dataset. We employed an 80/20 random split of the dataset to obtain training vs. test data. Dataset characteristics are shown in Table 1.

---

[5]Implementation and training is detailed in Appendix B.

To generate labels for timing and congestion, we use RTL Architect from Synopsys [13], transforming all designs to their physical layout, using the open-source *Nangate 45nm* standard-cell library. We ran all our designs with an aggressive timing constraint of `0.25` ns, because Synopsys RTL-A only reports WNS when a timing constraints are actually violated.

Table 1: Characteristics of OpenABCD and extracted Verilog. Designs have between 300–30K LOC.

| Design | # of Modules | # of Lines | Design | # of Modules | # of Lines |
|---|---|---|---|---|---|
| aes | 2 | 301 | coyote | 114 | 176279 |
| ariane | 39 | 214930 | dynamic_node | 9 | 796 |
| black_parrot | 88 | 62948 | ethmac | 10 | 1168 |
| bp_be_top | 38 | 13289 | jpeg | 5 | 669 |
| bp_fe_top | 15 | 7363 | microwatt | 31 | 26033 |
| bp_multi_top | 89 | 32959 | swerv_wrapper | 57 | 16496 |
| bp_quad | 252 | 293281 | vanilla5 | 39 | 11577 |

**Hyperparameter Setting.** For congestion and timing detection, we employed XGBoost [19], Light-GBM [20], and an FNN, each tuned to handle class imbalance and optimize predictive performance. XG-Boost was configured with default hyperparameters: `scale_pos_weight` set as the ratio of the majority to minority class to mitigate imbalance, `max_depth=30`, `learning_rate=0.05` and `n_estimators=500`. LightGBM used `is_unbalance=True` for automatic class weight adjustment, and default settings of `num_leaves=100`, `learning_rate=0.05`, and `feature_fraction=0.8`. The regression head followed a similar training procedure using the 'XGBRegressor' with a squared error loss. The FNN consisted of a single sigmoid neuron trained with binary cross-entropy (BCE) loss and was optimized using Adam with a learning rate of $1e^{-4}$.

**Metrics.** For congestion and timing classification tasks, we use the **F1-score**, **precision**, and **recall** to measure the balance between sensitivity and specificity. For the regression task of WNS prediction, we employ **R²** and **mean absolute percentage error (MAPE)**, providing insights for goodness-of-fit and prediction error relative to the target.

**Hardware.** *CL-Verilog* feature extraction was performed on a single NVidia H100, and downstream classifiers (XGBoost and LightGBM) were trained/evaluated on a CPU machine with 32GB RAM and 8 CPU cores. The FNN model was trained/evaluated using an NVidia RTX 8000 GPU.

## 4.2   Line-level Classification Results

We begin by discussing `VeriLoC`'s performance on line-level classification for both congestion and timing prediction. Table 2 tabulates our results for three different classification heads, as well as different context lengths ($p = \{1, 3, 5\}$).

**Congestion Detection.** As shown in Table 2, the highest F1-score for congestion detection is **0.86**, achieved by the LightGBM classifier with a context length of 5. This result underscores the importance of contextual information in accurately identifying congestion-causing lines.

**Timing Detection.** The highest F1-score of **0.95** is obtained using the XGBoost and LightGBM classifiers with a context length of 5, similar to the best performing model for congestion detection. Interestingly, timing prediction results are less sensitive to local context compared to congestion prediction. We achieve F1-scores of 0.83 for timing prediction even without local context ($p = 0$).

These results reinforce prior observations about the advantage of XGBoost and LightGBM over deep networks in handling imbalanced datasets and irregular feature distributions [56], albeit these results were in the context of tabular data. XGBoost and LightGBM improve F1-scores from 0.77 to 0.86 for congestion prediction and 0.83 to 0.95 for timing prediction.

Table 2: Performance of `VeriLoC` on line-level congestion and timing detection. Best results are highlighted in blue.

| Classifier | Context Length | Congestion | | | Timing | | |
|---|---|---|---|---|---|---|---|
| | | P | R | F1 | P | R | F1 |
| FNN | 0 | 0.38 | 0.74 | 0.50 | 0.67 | 0.88 | 0.76 |
| | 3 | 0.41 | 0.77 | 0.54 | 0.71 | 0.89 | 0.80 |
| | 5 | 0.86 | 0.70 | 0.77 | 0.76 | 0.92 | 0.83 |
| XGB | 0 | 0.38 | 0.74 | 0.50 | 0.76 | 0.92 | 0.83 |
| | 3 | 0.42 | 0.78 | 0.55 | 0.91 | 0.93 | 0.92 |
| | 5 | 0.94 | 0.78 | 0.85 | 0.94 | 0.94 | 0.94 |
| LGBM | 0 | 0.38 | 0.78 | 0.51 | 0.82 | 0.91 | 0.83 |
| | 3 | 0.41 | 0.76 | 0.53 | 0.93 | 0.92 | 0.92 |
| | 5 | 0.94 | 0.79 | 0.86 | 0.96 | 0.94 | 0.95 |

As shown in Table 2, they consistently outperform FNNs in both congestion and timing detection, particularly with a larger context of 5. The highest F1-scores for congestion detection (**0.86**) and timing detection (**0.95**) were achieved by these models, reinforcing their robustness in handling

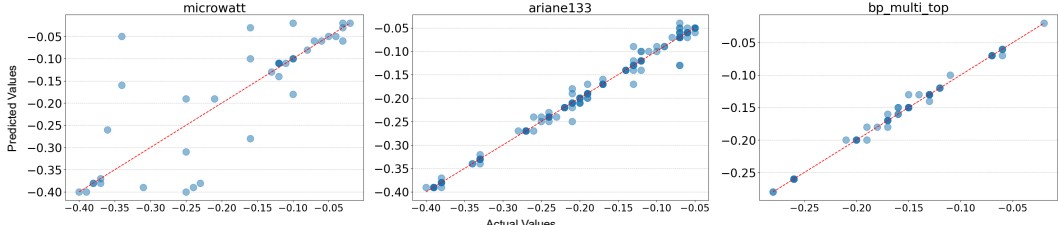

Figure 4: Scatter plots of actual vs. predicted line-level WNS using `VeriLoC` for three Verilog projects. In most instances, the predictions follow the actual WNS closely.

imbalanced datasets. The computational efficiency of XGBoost and its robustness in optimizing for minority class representation contributed to its superior performance compared to FNNs.

### 4.3 Timing Prediction and Comaprisons with SoTA

Table 3: Line-level prediction of WNS using `VeriLoC`. Module level prediction obtained from line-level ones.

| Design | R² | MAPE | Design | R² | MAPE |
|---|---|---|---|---|---|
| aes | 0.97 | 0.03 | coyote | 0.99 | 0.03 |
| ariane | 0.96 | 0.06 | dynamic_node | 0.76 | 0.18 |
| black_parrot | 0.99 | 0.01 | ethmac | 0.96 | 0.05 |
| bp_be_top | 0.76 | 0.27 | jpeg | 0.99 | 0.07 |
| bp_fe_top | 0.99 | 0.02 | microwatt | 0.93 | 0.21 |
| bp_multi_top | 0.99 | 0.02 | swerv_wrapper | 0.94 | 0.08 |
| bp_quad | 0.98 | 0.02 | vanilla5 | 0.99 | 0.02 |

Table 3 reports `VeriLoC`'s performance for predicting WNS at the line-level. Across all designs, `VeriLoC`'s achieves high $R^2$ and low MAPE across all designs but two. The accuracy of our WNS predictions are also evident Figure 4, where we compare actual vs. predicted WNS for all lines in three benchmarks.

We now compare `VeriLoC`'s module-level WNS prediction against state-of-art approaches. Recall that `VeriLoC`'s module-level WNS predictions are obtained by *first* predicting WNS of each line in the code and then picking the smallest WNS, while other methods only use entire module-level features. Table 4 compares `VeriLoC` with MasterRTL [4] and RTL-Timer [1], SoTA methods that use handcrafted features derived from RTL. Across designs, `VeriLoC` outperforms prior art, with higher R² values (always >0.90) and much lower MAPE (always <0.10).

Table 4: `VeriLoC` vs. SOTA on module-level WNS (timing) prediction. `VeriLoC` substantially improves R² and MAPE. Best results are in blue.

| Design | MasterRTL | | RTL-Timer | | VeriLoC | |
|---|---|---|---|---|---|---|
| | R² | MAPE | R² | MAPE | R² | MAPE |
| aes | 0.67 | 0.26 | 0.76 | 0.23 | 0.97 | 0.08 |
| ariane | 0.71 | 0.15 | 0.79 | 0.11 | 0.93 | 0.07 |
| black_parrot | 0.75 | 0.10 | 0.83 | 0.07 | 0.99 | 0.02 |
| bp_be_top | 0.76 | 0.12 | 0.87 | 0.08 | 0.98 | 0.08 |
| bp_fe_top | 0.80 | 0.09 | 0.88 | 0.05 | 0.98 | 0.04 |
| bp_multi_top | 0.61 | 0.16 | 0.67 | 0.13 | 0.99 | 0.02 |
| bp_quad | 0.69 | 0.14 | 0.78 | 0.10 | 0.94 | 0.06 |
| coyote | 0.74 | 0.15 | 0.83 | 0.12 | 0.99 | 0.03 |
| dynamic_node | 0.73 | 0.20 | 0.80 | 0.16 | 0.99 | 0.06 |
| ethmac | 0.77 | 0.24 | 0.83 | 0.21 | 0.99 | 0.03 |
| jpeg | 0.82 | 0.29 | 0.90 | 0.26 | 0.98 | 0.04 |
| microwatt | 0.81 | 0.20 | 0.80 | 0.16 | 0.99 | 0.02 |
| swerv_wrapper | 0.69 | 0.27 | 0.76 | 0.24 | 0.95 | 0.07 |
| vanilla5 | 0.68 | 0.19 | 0.74 | 0.16 | 0.98 | 0.07 |

Table 5 compares `VeriLoC` with three additional baselines on module-level WNS: GNN-based predictors from synthesized netlists [10], VeriDistill [11] that uses LLM-based RTL embeddings with GNN-based synthesized look-up-table (LUT) embeddings, and as an ablation, `VeriLoC`-mod, a version of `VeriLoC` (`VeriLoC`-mod) that only uses module-level but no line-level embeddings. `VeriLoC` achieves the best performance, notably improving upon `VeriLoC`-mod, demonstrating the value of line-level embeddings even for module-level timing prediction. VeriDistill is second on MAPE, but performs poorly on $R^2$.

Direct comparisons of `VeriLoC` with *CircuitFusion* [12] method, a very recent, state-of-art, module-level multimodal PPA predictor, are challenging because they use a different dataset. Still, when comparing the respective improvements upon RTL-Timer, we see that *CircuitFusion* reports an increase in R² from 0.81 → 0.83 and a reduction in MAPE from 16% → 11%, whereas `VeriLoC` demonstrates larger relative gains, with R² increasing from 0.86 → 0.94, and MAPE reducing from 12% → 6%. We caution against reading more

Table 5: Comparison with SOTA methods for timing prediction on the OpenABCD benchmark. `VeriLoC`-mod uses *only* module embeddings but no line embeddings.

| Metric | GNN [10] | MasterRTL [4] | RTL-Timer [1] | VeriDistill [11] | VeriLoC-Mod | VeriLoC |
|--------|----------|---------------|---------------|------------------|-------------|---------|
| $R^2$  | 0.69     | 0.74          | 0.86          | 0.728            | 0.91        | **0.94** |
| MAPE   | 0.17     | 0.15          | 0.12          | 0.076            | 0.08        | **0.06** |

into these comparisons because they are on different datasets, and also emphasize that `VeriLoC`'s primary goal of LoC-level predictions is different from *CircuitFusion*'s.

## 4.4 Discussion

We now comment on some interesting properties of `VeriLoC`, avenues for future improvement, and alternate baselines.

**Runtime Comparisons.** We compare the runtime efficiency of `VeriLoC` over Synopsys RTL Architect [13] (our synthesis and PnR tool)—using *CL-Verilog*-13B as its base, `VeriLoC` achieves $14\times$ average speedup and a median speedup of $61\times$. To explore the potential for even greater runtime improvement, we trained a 7B *CL-Verilog* model using the training procedure described in [14]. The 7B model has a $22\times$ on average and $113\times$ median speedup with a modest tradeoff in F1 score of 0.93 for timing and 0.84 for congestion. Thus, while the focus of `VeriLoC` is on accurate line-level PPA prediction, smaller models like the 7B variant or Mixture-of-Experts (MoE) based architectures can be adopted to further prioritize inference speed without significant degradation in predictive accuracy.

**Impact of Verilog Code Length on Accuracy.** To assess the effect of Verilog file length on model performance, we conducted a stratified analysis by splitting test samples based on the number of lines per file. We observed a negligible degradation in performance for longer inputs. Specifically, the **congestion F1-score** dropped slightly from **0.862** for files with fewer than 2000 lines to **0.855** for files with more than 2000 lines. Similarly, the **timing F1-score** decreased from **0.953** to **0.946**, and the **Mean Absolute Percentage Error (MAPE)** increased from **5%** to **6.5%**. These results indicate that the model exhibits strong robustness to variations in input length, with only minor performance degradation observed on longer files.

**Can Black-Box LLMs Predict Hardware Design Quality?** As we have noted before, much of the work in the software community on line-level bug detection uses prompt engineering with pre-trained models. These strategies have been successful because of the abundance of training data in the software world. However, hardware data is scarce, especially so for complex concepts like routing congestion. Thus, we hypothesize that prompting strategies are unlikely to work in the hardware context for line-level detection of congestion and timing issues. To test this hypothesis, we picked 10 Verilog files from our test dataset and asked ChatGPT-4o [57] to identify lines of code that would cause timing or congestion issues. ChatGPT was unable to identify lines responsible for congestion in *any* of our trials. For timing, ChatGPT identified a line of code correctly in one instance, but also had a large number of false positives. Examples of chat responses are given in Appendix D (Fig. 6–9).

## 5 Conclusion

This paper presents a novel LLM-driven framework, `VeriLoC`, which provides real-time feedback to hardware designers for the impact of their RTL code on crucial design-quality metrics, timing and congestion. By leveraging embedding from LLMs, `VeriLoC` bridges the gap between RTL coding and downstream performance evaluation, enabling designers to make informed decisions early in the design process. Results demonstrate `VeriLoC`'s capability in predicting design metrics at RTL stage, both for individual lines of code and at the module level.

**Limitations.** Despite its strong performance, VeriLoC has several important limitations. First, it currently operates on leaf modules drawn from the Open-ABCD benchmark and has not been adapted to large-scale industrial designs with inter-module dependencies. Extensive validation on diverse, industry-grade RTL corpora will establish robustness and generalization, but is hindered by open access to such data.

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

# A    IC Design

The design of modern ICs with billions of transistors and ever-more complex technology nodes is highly demanding. Thus, sophisticated tooling is essential in this domain known as electronic design automation (EDA). EDA tools are build upon strong notions of abstractions and hierarchical procedures. Next, we outline the industry-standard approach for these procedures. More details relevant for this work are also discussed in subsections in here.

System Specification, Architectural Design. Objectives and requirements for functionality, performance, and physical implementation are formulated. Modeling languages like SystemC can be used for a formalized approach.

Behavioral and Logic Design. The specification and architecture description are transformed into a behavioral model which describes inputs, outputs, timing behavior, etc. for the whole system. Toward that end, specific hardware description languages (HDLs) like Verilog are utilized. The abstraction level for this process is also often referred to as register-transfer level (RTL). To limit design time and efforts, third-party components, so-called IP modules, can be integrated at this stage.

Logic Synthesis. The behavioral model is transformed into a low-level circuit description, the gate-level netlist (GLN). This step requires a technology library for mapping from a generic circuit to the technology-specific circuit that is to be manufactured in the end.

Physical Design. The GLN is transformed into an actual physical layout of gates, memories, interconnects, etc. Given the high complexity of this stage, it is typically further divided into the following tasks: partitioning and/or floorplanning, power and ground delivery, placement, clock delivery, routing, and timing closure.

Verification and Signoff. The physical layout must be verified against various design and manufacturing rules, to ensure correct functionality and electrical behaviour. Once all rules are met, the design can be signed-off and taped-out, i.e., send out for fabrication, packaging, and testing.

**Practical Challenge.** Despite the general success of this compartmentalized approach, a key challenge remains: going through the full EDA stack end-to-end takes considerable time and efforts, in the range of months even for large teams. This challenge is know as the "productivity gap" and is expected to stay, especially for ever-more advanced technology nodes [58]. In an effort for best design quality, engineers often need to reiterate many times over key processes like placement and routing. Due to the very nature of the hierarchical tooling, the individual processes are often lacking detailed insights from prior stages and, more concerning, reasonable estimates for their impact on processes further down in the pipeline. In other words, there is a strong need to integrate well-informed quality assessment into early stages of the EDA pipeline—this reiterates the main motivation of this work at hand.

# B    Encoder-Decoder Architecture for Dimensionality Reduction

This model is designed to effectively compress high-dimensional embeddings into a compact latent space while preserving their semantic content. Below, we analyze its structure and key features:

## B.1    Network Architecture

The encoder-decoder network consists of fully connected layers with batch normalization, dropout, and non-linear activation functions. The detailed architecture is shown in Table 6.

## B.2    Regularization Techniques

To ensure robust learning and prevent overfitting, the following techniques are incorporated:

- **Dropout:** Applied with a 30% probability at multiple layers to prevent co-adaptation of neurons.
- **Batch Normalization:** Normalizes activations at each layer to stabilize training and improve convergence.
- **LeakyReLU:** Chosen over standard ReLU to avoid dead neurons and ensure a small gradient for negative values.

Table 6: Detailed architecture of the encoder-decoder network.

| Layer Type | Number of Units | Activation Function | Additional Features |
|---|---|---|---|
| **Encoder** | | | |
| Fully Connected | 4096 | LeakyReLU ($\alpha = 0.01$) | BatchNorm, Dropout (30%) |
| Fully Connected | 1024 | LeakyReLU ($\alpha = 0.01$) | BatchNorm, Dropout (30%) |
| Fully Connected | Latent Dim ($d$) | LeakyReLU ($\alpha = 0.01$) | BatchNorm |
| **Decoder** | | | |
| Fully Connected | 1024 | LeakyReLU ($\alpha = 0.01$) | BatchNorm, Dropout (30%) |
| Fully Connected | 4096 | LeakyReLU ($\alpha = 0.01$) | BatchNorm, Dropout (30%) |
| Fully Connected | Input Dim | - | - |

## B.3 Optimization Strategy

The model is trained using the `AdamW` optimizer, which combines adaptive learning rates with weight decay regularization. Additional details include:

- **Learning Rate:** Set to a small value ($10^{-4}$) to ensure gradual convergence.
- **Reconstruction Loss:** The mean squared error (MSE) between the input embeddings and their reconstructed versions is minimized.
- **Weight Initialization:** All linear layers are initialized using `Xavier Uniform Initialization` to ensure balanced gradients at the start of training.

## B.4 Training Procedure

The model is trained over 200 epochs using mini-batch gradient descent, with the following considerations:

- **Batch Size:** A batch size of 128 is chosen to balance memory efficiency and gradient stability.
- **Orthogonality Regularization:** Although not implemented in this iteration, orthogonality constraints on the encoder weights can further enhance disentanglement in the latent space.
- **Validation:** Validation loss on a separate test dataset is monitored to ensure the model generalizes to unseen data.

## B.5 Usage in Downstream Tasks

The trained encoder produces latent embeddings that serve as input features for classification and regression tasks. These embeddings are compact, noise-robust, and retain essential semantic information from the original input.

## B.6 Justification for Using the Last Hidden Layer in `VeriLoC`

Although `VeriLoC` focuses on line-level quality prediction, we first conducted a complementary ablation study on module-level embeddings to evaluate the predictive quality of hidden states extracted from different layers of the CL-Verilog model. As shown in Table 7, we observed that embeddings derived from the *last decoder layer* consistently achieved the highest accuracy across all quality-of-result (QoR) metrics—yielding the best $R^2$ scores and the lowest MAPE for both timing and congestion prediction. These results suggest that the final layer best captures high-level semantic and structural information necessary for downstream design-quality tasks.

Based on this empirical evidence, `VeriLoC` exclusively uses the last hidden layer to extract both line-level and module-level embeddings. This design choice ensures that the model benefits from the richest representation available, without introducing ambiguity or requiring additional architectural tuning to select among intermediate layers. Furthermore, the consistently superior performance of the final layer justifies avoiding ensemble or multi-layer fusion strategies, which may add unnecessary complexity without proportional gains.

Table 7: Prediction quality using embeddings derived from hidden states of the model, specifically examining the first (1st, 2nd, 3rd) and last (3rd-Last, 2nd-Last, Last) layers for *CL-Verilog*.

| Hidden Layer | Timing | | Congestion | |
|:---:|:---:|:---:|:---:|:---:|
| | R² | MAPE | R² | MAPE |
| 1st | 0.66 | 15.46 | 0.66 | 18.13 |
| 2nd | 0.77 | 5.64 | 0.59 | 23.92 |
| 3rd | 0.75 | 6.55 | 0.72 | 13.62 |
| 3rd-Last | 0.85 | 5.88 | 0.53 | 22.34 |
| 2nd-Last | 0.85 | 9.32 | 0.59 | 17.56 |
| Last | 0.89 | 5.57 | 0.74 | 11.66 |

Table 8: Impact of Hidden Dimension and Classifier on Line-Level Detection Performance for Congestion and Timing. Precision (P), Recall (R).

| | Hidden Dim. | Congestion | | Timing | |
|:---:|:---:|:---:|:---:|:---:|:---:|
| | | P | R | P | R |
| XGB | 32 | 0.76 | 0.64 | 0.77 | 0.71 |
| | 64 | 0.88 | 0.71 | 0.86 | 0.83 |
| | 128 | 0.94 | 0.78 | 0.94 | 0.94 |
| | 256 | 0.94 | 0.78 | 0.94 | 0.94 |
| LGBM | 32 | 0.77 | 0.64 | 0.79 | 0.71 |
| | 64 | 0.88 | 0.72 | 0.86 | 0.82 |
| | 128 | 0.94 | 0.79 | 0.96 | 0.94 |
| | 256 | 0.94 | 0.79 | 0.96 | 0.94 |

## C  Additional Ablation Studies

### C.1  Role of Encoder and Choice of Hidden Dimensions

As noted in Section 3.3, we use an encoder-decoder architecture to reduce the dimensionality of raw line- and module-level embeddings before classification/regression. Without the encoder, we obtained F1-scores of less than 0.5. To determine the optimal hidden dimension for the encoder, we study different embedding dimensions across XGBoost and LightGBM. The results are summarized in Table 8.

We find that increasing the hidden dimension significantly improves both congestion and timing detection performance up to a dimension of 128. Beyond 128, however, there is no observable gain in performance, suggesting that further increasing the embedding size is not warranted and that 128 is the optimal hidden dimension, balancing performance and computational efficiency.

### C.2  Impact of Comments in RTL

As opposed to intermediate representations like AIGs, RTL code also contains comments that we hypothesized to be useful in design quality predictions. Here, we evaluate the role of comments in VeriLoC's accuracy. To this end, we conducted a study where we removed comments from both the train and test datasets and train VeriLoC classifiers on the resulting code. Table 9 compares line-level detection performance with (w) vs without (w/o) comments in the embeddings. The results show a modest but clear improvement in congestion and timing precision, recall, and F1-scores when comments are included. The benefits are starkest for FNN, where F1-scores increase from 0.71 to 0.77 for timing and 0.70 to 0.76 for congestion. XGBoost and LightGBM also benefit from comments, especially for congestion prediction with F1-scores increasing from 0.92 to 0.95 in the best case.

Table 9: Effect of Comments (C) in Module Embedding Generation on Line Detection Performance. Precision (P), Recall (R).

| | Emb. | Congestion | | | Timing | | |
|---|---|---|---|---|---|---|---|
| | | P | R | F1 | P | R | F1 |
| FNN | w/o C | 0.80 | 0.64 | 0.71 | 0.63 | 0.78 | 0.70 |
| | w/ C | 0.86 | 0.7 | 0.77 | 0.67 | 0.88 | 0.76 |
| XGB | w/o C | 0.93 | 0.76 | 0.84 | 0.91 | 0.92 | 0.91 |
| | w/ C | 0.94 | 0.78 | 0.85 | 0.94 | 0.94 | 0.94 |
| LGBM | w/o C | 0.95 | 0.77 | 0.85 | 0.93 | 0.91 | 0.92 |
| | w/ C | 0.94 | 0.79 | 0.86 | 0.96 | 0.94 | 0.95 |

Fig. 5 further supports these findings through saliency-based visualizations of batched vs individual line embeddings. Specifically, the saliency heatmaps reveal that, w/o comments, key structural elements like `always @(posedge clk)` receive a disproportionate amount of attention, whereas w/ comments, the model distributes attention more effectively across relevant components.

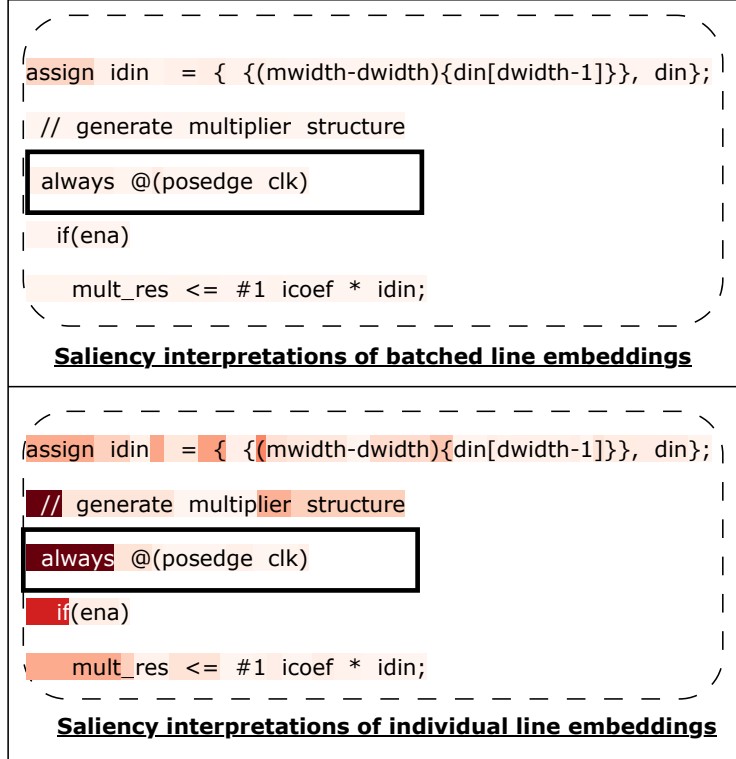

Figure 5: Saliency map comparing attention of `VeriLoC` model (bottom) vs. batched line embeddings (top).

## C.3 Alternate Approaches

Recall that including neighboring lines of code can significantly improve performance, especially for congestion prediction. An alternate method for including local context, however, would be to obtain a single embedding for a batch of consecutive lines $B = \{l_{i-p}, \ldots, l_i, \ldots, l_{i+p}\}$ by passing

$B$ through *CL-Verilog* as a single input. The model computes batched embeddings $z_{\text{batch}}(B)$, which natively capture inter-line dependencies via the LLM's attention mechanism.

Although appealing, the batched approach results in lower F1-scores, achieving at best an F1-score of 0.8 for congestion prediction; recall that `VeriLoC` achieves an F1-score of 0.86. Fig. 5 depicts the attention patterns of the model for congestion detection where the line `always @(posedge clk)` is critical. For the batched approach, the model's attention is dispersed across multiple unrelated tokens. Conversely, line-wise embeddings allow the model to prioritize the relevant tokens more effectively, as shown by the darker-red shades. This also aligns with the findings in [59], which emphasize the importance of modular embedding strategies for structured tasks.

## D   Case Study Using `GPT-4o` and `VeriLoC`

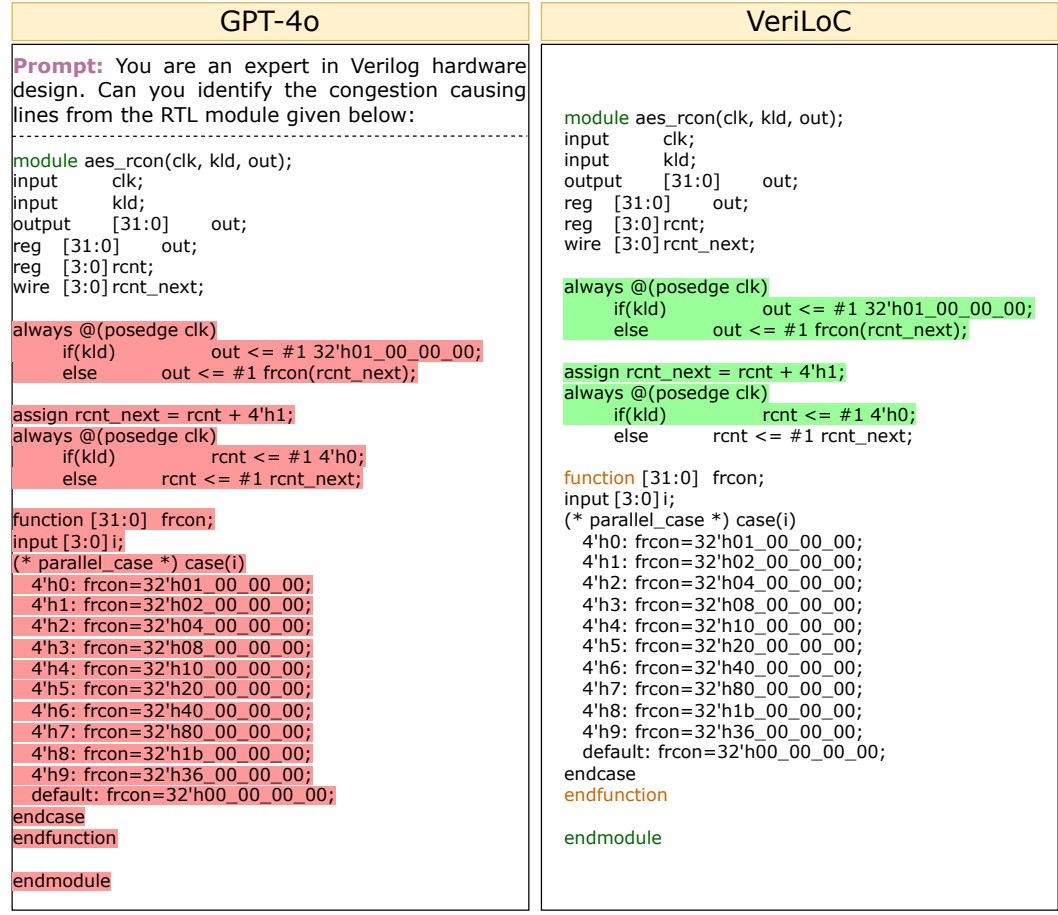

Figure 6: The lines of code highlighted by GPT-4o and `VeriLoC` for congestion metrics on `aes_rcon` design.

As a part of the case study, to understand the capability of generic LLMs to analyze RTL code, we took `GPT-4o` [57] and prompted it to report the line numbers in the code responsible for timing and congestion, respectively. We also showcase `VeriLoC` performance for the same codes. `GPT-4o` was unable to detect the correct line numbers, i.e, showed false positives, whereas `VeriLoC` has shown significant results when compared to the ground truth obtained using the EDA tool.

Fig. 6–9 show such examples with highlighted text being the line number reported by `GPT-4o` and `VeriLoC`, respectively. Fig. 6 refers to the `aes_rcon` design, where `GPT-4o` predicts every line as critical for congestion issues, whereas `VeriLoC` selectively highlights the correct lines related to congestion. Fig. 7 again shows an inability of `GPT-4o`, now for the task of timing prediction.

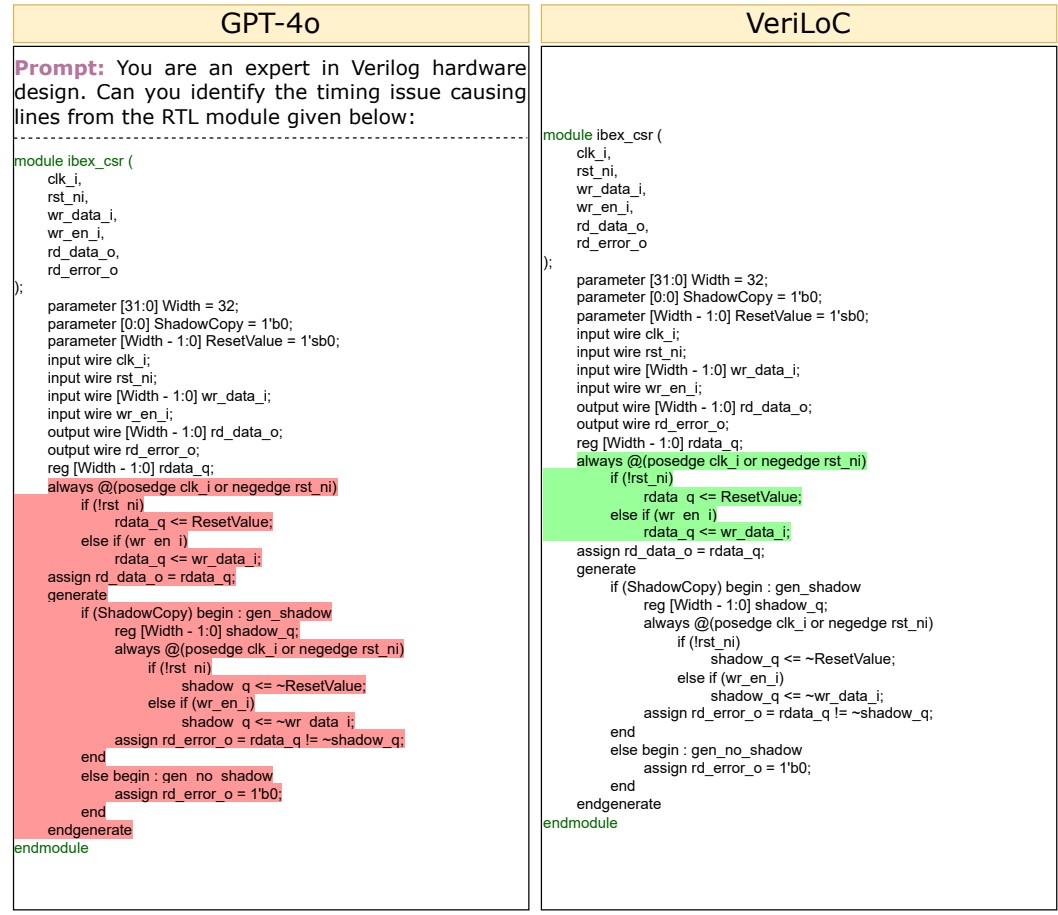

Figure 7: The lines of code highlighted by GPT-4o and `VeriLoC` for timing metrics on `ibex_csr` design.

Although it has highlighted the lines related to timing impact, namely the first `always` block, it also reports the next blocks as potential candidates. In Fig. 8, `GPT-4o` picks `registers` definitions as a potential reason for congestion and skips the actual criticial line containing the additions operation. Finally, in Fig. 9 `GPT-4o` was unable to pick the correct `always` block responsible for timing issues.

| GPT-4o | VeriLoC |
|---|---|

**GPT-4o**

**Prompt:** You are an expert in Verilog hardware design. Can you identify the congestion causing lines from the RTL module given below:

- - - - - - - - - - - - - - - - - - - - - - - - - - - - - -

```
wire [mwidth-1:0] idin;
wire [mwidth-1:0] icoef;

reg  [mwidth -1:0] mult_res;
wire [rwidth -1:0] ext_mult_res;

//
// module body
//
assign icoef = { {(mwidth-cwidth){coef[cwidth-1]}}, coef};
assign idin  = { {(mwidth-dwidth){din[dwidth-1]}}, din};

// generate multiplier structure
always @(posedge clk)
    if(ena)
        mult_res <= #1 icoef * idin;

assign ext_mult_res = { {3{mult_res[mwidth-1]}}, mult_res};

// generate adder structure
always @(posedge clk)
   if(ena)
        if(dclr)
            result <= #1 ext_mult_res;
        else
            result <= #1 ext_mult_res + result;
```

**VeriLoC**

```
wire [mwidth-1:0] idin;
wire [mwidth-1:0] icoef;

reg  [mwidth -1:0] mult_res;
wire [rwidth -1:0] ext_mult_res;

//
// module body
//
assign icoef = { {(mwidth-cwidth){coef[cwidth-1]}}, coef};
assign idin  = { {(mwidth-dwidth){din[dwidth-1]}}, din};

// generate multiplier structure
always @(posedge clk)
    if(ena)
        mult_res <= #1 icoef * idin;

assign ext_mult_res = { {3{mult_res[mwidth-1]}}, mult_res};

// generate adder structure
always @(posedge clk)
   if(ena)
        if(dclr)
            result <= #1 ext_mult_res;
        else
            result <= #1 ext_mult_res + result;
```

Figure 8: The lines of code highlighted by GPT-4o and `VeriLoC` for congestion metrics on `dct_mac` design.

| GPT-4o | VeriLoC |
|---|---|

**GPT-4o**

**Prompt:** You are an expert in Verilog hardware design. Can you identify the timing issue causing lines from the RTL module given below:

```verilog
    output [11:0] dout;
    output      douten; // data-out enable

    //
    // variables
    //

    reg ld_zigzag;
    reg [11:0] sresult [63:0]; // store results for zig-zagging
    //
    // module body
    //

    always @(posedge clk)
     if(ena)
        ld_zigzag <= #1 dstrb;

    assign douten = ld_zigzag;

    integer n;

    always @(posedge clk)
     if(ena)
       if(ld_zigzag)   // reload results-register file
        begin
          sresult[63] <= #1 din_00;
          sresult[62] <= #1 din_01;
                .
                .
                .
                .
                .
                .
          sresult[09] <= #1 din_47;
          sresult[08] <= #1 din_56;
          sresult[03] <= #1 din_57;
          sresult[02] <= #1 din_67;
          sresult[01] <= #1 din_76;
        end
     else      // shift results out
       for (n=1; n<=63; n=n+1) // do not change sresult[0]
         sresult[n] <= #1 sresult[n -1];

    assign dout = sresult[63];
endmodule
```

**VeriLoC**

```verilog
    //
    // module body
    //

    always @(posedge clk)
     if(ena)
        ld_zigzag <= #1 dstrb;

    assign douten = ld_zigzag;
    integer n;

    always @(posedge clk)
     if(ena)
       if(ld_zigzag)   // reload results-register file
        begin
          sresult[63] <= #1 din_00;
                .
                .
                .
          sresult[45] <= #1 din_32;
          sresult[44] <= #1 din_41;
          sresult[43] <= #1 din_50;
                .
                .
                .
          sresult[33] <= #1 din_25;
          sresult[32] <= #1 din_34;
          sresult[31] <= #1 din_43;
          sresult[30] <= #1 din_52;
          sresult[27] <= #1 din_71;
          sresult[26] <= #1 din_62;
          sresult[19] <= #1 din_36;
          sresult[18] <= #1 din_45;
          sresult[17] <= #1 din_54;
          sresult[16] <= #1 din_63;
          sresult[15] <= #1 din_72;
          sresult[14] <= #1 din_73;
          sresult[13] <= #1 din_64;
          sresult[12] <= #1 din_55;
          sresult[11] <= #1 din_46;
          sresult[10] <= #1 din_37;
          sresult[09] <= #1 din_47;
          sresult[08] <= #1 din_56;
          sresult[03] <= #1 din_57;
          sresult[02] <= #1 din_67;
          sresult[01] <= #1 din_76;
        end
     else      // shift results out
       for (n=1; n<=63; n=n+1) // do not change sresult[0]
         sresult[n] <= #1 sresult[n -1];

    assign dout = sresult[63];
endmodule
```

Figure 9: The lines of code highlighted by GPT-4o and VeriLoC for timing metrics on zigzag design.

