# OpenReview forum: "VeriLoC: Line-of-Code Level Prediction of Hardware Design Quality from Verilog Code"
_NeurIPS.cc/2025/Conference — NeurIPS 2025 poster_

### Official Review · Reviewer_a3PJ · 2025-06-27

**Clarity:** 4
**Significance:** 3
**Originality:** 3
**Rating:** 5
**Confidence:** 4

**Summary:**

This paper proposes a method for predicting hardware design quality metrics (timing violations and routing congestion) by combining LLM-based Verilog code embeddings with tree-based models (XGBoost/LightGBM).

**Questions:**

- How does the model perform on designs with different coding styles (e.g., generated RTL vs. handwritten)?
- Are there plans to integrate this into industry workflows (e.g., as a plugin for Vivado/VSCode)? What are the inference latency and memory overheads?
- How would NeurIPS audiences be interested in this? I honestly suggest to submit to EDA venues (DAC, DATE, ICCAD, etc.).

**Ethical Concerns:**

["NO or VERY MINOR ethics concerns only"]

**Final Justification:**

I would like to keep my positive assessment. This is a good paper discussing on a valuable and interesting problem. The method is a bit flat in flavor but that is ok. I do acknowledge the alignment of topics after rebuttal.

**Limitations:**

yes

**Quality:**

3

**Strengths And Weaknesses:**

Strengths:
- The paper is well-structured, and figures (e.g., pipeline overview) effectively illustrate the workflow.
- The methodology is technically sound, with thorough ablation studies (e.g., dimensionality reduction, embedding granularity) supporting design choices.
- Addresses a real-world pain point in RTL design verification, potentially reducing EDA tool runtime.

Weaknesses:
- Straightforward application of embedding LLM. LLM + dimensionality reduction + classifier is not fundamentally novel outside hardware contexts, adding little value to the field of neural information processing.

---

> ### Author Rebuttal · Authors · 2025-07-30
>
> > `Q1` Straightforward application of embedding LLM. LLM + dimensionality reduction + classifier is not fundamentally novel outside hardware contexts, adding little value to the field of neural information processing.
>
> From a machine learning perspective, we believe that our *VeriLoC architecture is the first to blend both line-level and module-level embeddings for line-level classification*, and as such *might also be useful for line-of-code level software prediction tasks* (e.g., line-level bug prediction/dectection). We bring in new observations including the role of local context around each line of code and its importance in achieving state-of-art performance. There is also a growing body of work at ML conference on ML/AI for hardware design (please see some references below) since it is an area where ML/AI is having major impact, and we hope to contribute to this body of work.
>
> ---
>
> > `Q2` How does the model perform on designs with different coding styles (e.g., generated RTL vs. handwritten)?
>
> Thanks for the excellent question. Our experiments thus far are on human-written RTL code, but could be replicated on generated RTL, for example, from a high-level synthesis tool. While we could not run these experiments in the rebuttal timeframe, we will add this as a topic for further study.
>
> ---
>
> > `Q3` Are there plans to integrate this into industry workflows (e.g., as a plugin for Vivado/VSCode)? What are the inference latency and memory overheads?
>
> Thanks for the great question! Yes, we are investigating integration into industry workflows. On average, inference takes ~5.6 seconds per Verilog module when using the 13B CL-Verilog model. The end-to-end process, including embedding extraction, encoding, and prediction, requires up to 26 GB of GPU memory. We also trained a 7B CL-Verilog variant, which reduces memory usage and latency to 3.5 seconds with only minor accuracy tradeoffs (see Section 4.4).
>
> ---
>
> > `Q4` How would NeurIPS audiences be interested in this? I honestly suggest to submit to EDA venues (DAC, DATE, ICCAD, etc.).
>
> There is a growing body of work published in leading AI/ML conferences on AI/ML for hardware design and the novel challenges this application domain brings (see references [1]-[5] below as examples).  The CircuitFusion paper on module-level timing prediction was also recently published at ICLR’25.  As such, we do think NuerIPS attendees would be interested in this topic of growing relevance. Furthermore, the proposed VeriLoC architecture might be useful in software prediction tasks like line-level bug detection, and therefore, we might have interest from the "AI for software coding" community as well.
>
>
> ---
>
> `References:`
>
> [1] Wenji Fang, Shang Liu, Jing Wang, & Zhiyao Xie (2025). CircuitFusion: Multimodal Circuit Representation Learning for Agile Chip Design. In The Thirteenth International Conference on Learning Representations.
>
> [2] Animesh Basak Chowdhury, Marco Romanelli, Benjamin Tan, Ramesh Karri, & Siddharth Garg (2024). Retrieval-Guided Reinforcement Learning for Boolean Circuit Minimization. In The Twelfth International Conference on Learning Representations.
>
> [3] Vasudevan, S., Jiang, W., Bieber, D., Singh, R., shojaei, h., Ho, C., & Sutton, C. (2021). Learning Semantic Representations to Verify Hardware Designs. In Advances in Neural Information Processing Systems (pp. 23491–23504). Curran Associates, Inc..
>
> [4] Zehua PEI, Huiling Zhen, Mingxuan Yuan, Yu Huang, & Bei Yu (2024). BetterV: Controlled Verilog Generation with Discriminative Guidance. In Forty-first International Conference on Machine Learning.
>
> [5] Yang, S., Yang, Z., Li, D., Zhang, Y., Zhang, Z., Song, G., & Hao, J. (2022). Versatile Multi-stage Graph Neural Network for Circuit Representation. In Advances in Neural Information Processing Systems (pp. 20313–20324). Curran Associates, Inc..

---

### Official Review · Reviewer_MVt5 · 2025-06-27

**Clarity:** 3
**Significance:** 3
**Originality:** 3
**Rating:** 4
**Confidence:** 3

**Summary:**

This paper proposes VeriLoC, a novel method for predicting timing and routing congestion directly from Verilog code at both the line and module levels. By leveraging LLM-based embeddings, VeriLoC achieves strong performance improvements over prior work. The approach aims to provide early-stage design-quality prediction and may be useful for other hardware optimization tasks.

**Questions:**

- Why were context windows larger than 5 not tested or reported in Table 2?
- The timing prediction task is evaluated using binary classification metrics. Could the authors clarify why binary classification was chosen for this task, and what the criteria are for defining positive and negative cases?

**Ethical Concerns:**

["NO or VERY MINOR ethics concerns only"]

**Final Justification:**

Thank you so much for addresssing my concerns. I will maintain my score. I hope the authors could add more experiment details as they promised.

**Limitations:**

yes

**Paper Formatting Concerns:**

There are too many embedded tables on page 8, which makes the layout somewhat confusing.

**Quality:**

3

**Strengths And Weaknesses:**

strength:
- A key strength of this work is its novel focus on predicting design quality at the line level of RTL code, which has not been explored in previous studies. The results also demonstrate that LLM-based embeddings can effectively capture the semantic information needed for accurate line-level quality prediction.
- The paper is very clearly written, making the methodology and results easy to understand.
- This work is that the authors have open-sourced their code, which will facilitate reproducibility and encourage further research in this area.

weakness:

- My main concern is the practical applicability of this approach, as directly predicting timing and congestion from RTL code alone is inherently inaccurate. In practice, both timing and congestion are heavily influenced by downstream logic synthesis and physical implementation algorithms, which are not reflected at the RTL level. Therefore, I also find the analysis in Figures 7-9 to be not entirely convincing or clearly justified.

---

> ### Author Rebuttal · Authors · 2025-07-30
>
> > `Q1` My main concern is the practical applicability of this approach, as directly predicting timing and congestion from RTL code alone is inherently inaccurate. In practice, both timing and congestion are heavily influenced by downstream logic synthesis and physical implementation algorithms, which are not reflected at the RTL level. Therefore, I also find the analysis in Figures 7-9 to be not entirely convincing or clearly justified.
>
> - We completely agree with the reviewer, and note that the impact of synthesis and physical design is implicitly captured by VeriLoC via its ground-truth labels that are obtained from an EDA tool like Synopsys RTL Architect that first performs full synthesis and place-and-route and then back-annotates timing slack and congestion metrics at the line-level. By training against these ground-truth labels, VeriLoC provides quick and early estimation at the line-level without having to go through this time-consuming design flow.
>
> - If the flow parameters or constraints change, these ground-truth labels will change also and the classifier/prediction layers of VeriLoC will be retrained. To illustrate this, we ran a new experiment by re-running the flow with stricter timing constraints, resulting in more LoCs with timing and congestion violations. We then retrained the classification/prediction layers on this new data and found that VeriLoC is still able to detect violations with high accuracy. We observe no performance drop with 0.84 F1 for congestion, 0.95 F1 for timing and 0.94 $R^2$ for WNS Prediction.
>
> ---
>
> > `Q2` Why were context windows larger than 5 not tested or reported in Table 2?
>
> - Thank you for the question. We did evaluate larger context windows, specifically p = 7, 9, and 11, during our ablation studies. However, we observed that increasing the window size beyond p = 5 did not lead to any improvement in F1-scores for either congestion or timing prediction. We will update the paper with additional details of these experiments.
>
> ---
>
> > `Q3` The timing prediction task is evaluated using binary classification metrics. Could the authors clarify why binary classification was chosen for this task, and what the criteria are for defining positive and negative cases?
>
> - Thank you for the question and apologies if this was not clear. The baseline commercial EDA tool that generates ground-truth labels only reports which lines exceed/violate a given timing or congestion constraint, since it is typical in industry to focus on focus on these parts of the design. Thus we set up a binary classification task where we assign a positive label to a line if its WNS is it violates a specified timing/congestion constraint and a negative label otherwise.
> - The tool also provides the actual worst negative slack (WNS) per line for lines that violate timing which we formulate as a regression problem and report results in Tables 3-5.

---

### Official Review · Reviewer_ryLD · 2025-07-02

**Clarity:** 2
**Significance:** 3
**Originality:** 3
**Rating:** 4
**Confidence:** 3

**Summary:**

This paper has proposed a workflow using llm to directly extract line-level and module-level information of verilog codes as a predection step to evaluate design quality at the very early stage of chip design. The objective is to predict metrics such as timing, congestion before all the time-consuming backend flows. In this work, the authors proposed using LLM as verilog encoder, followed with various line-level classifier/regressor module for different performance related task. The authors claims the finer line-level information and inter-line level information extractions contribute to higher metric prediciton/classification accuracy.

**Questions:**

1. As listed in the Weakness.
2. I also like to ask about the training cost, and the overall cost compared with baseline methods.
3. I want to know why you chose the exact model architecture?
4. Do you need to pretrain the encoder or not? Why do you call it LLM when it is just an  transformer model? unless it is pretrained on huge amount of general language data？

**Ethical Concerns:**

["NO or VERY MINOR ethics concerns only"]

**Final Justification:**

Two new experiments are added to by rerunning the flow with stricter timing constraints and evaluating a model with line-level encodings only without module encodings, which addressed weakness 1 & 2. The other 4 questions are well explained in the rebuttal as well.

**Limitations:**

yes

**Quality:**

3

**Strengths And Weaknesses:**

Strengths:
	1. The encoding strategy prosposed in this paper includes finer granularity. The concatenation of line-level rtl information and module-level information is richer than simple embedding of module rtl
	2. The paper has made a good point about hardware design data is very scarce publicly, thus very difficult to reply on pretrained LLM for direct performance prediction for the lack of hugh quality data in pretrain data. And the embedding stage + performance prediction/classification allowing for resource-limited training.
	3. The paper has shown sota performance prediction witht he help of finer rtl information over the previous baselines.

Weaknesses:
	1. The importance of each single line and how much will it affect the backend metrics such as congestion/timing is still beyond my imagination. For example, in Figure. 4, the author provided visualization of actual vs preducted WNS. But what if the backend EDA algorithms changes, will the strong alignment still hold? Such discussion or proof is missing, which will make the statement less convincing. For example, if the placement & routing algorithm changes or even if the technology nodes changes, will the WNS be the same? And for each line in the RTL, the code itself cleary did not change? How will the model adjust to these without additional information.
	2. A following concern is that The model did not provide ablative experiments on the line-level/module-level information. You need to provide it to prove the extra information does help to prediction the features bettern instead of just adopting a stronger model to fit the dataset better.
	3. Some more details about the training are not clear to me.  How exactly do you label the data of each line-level? How exacly do you train the encoder? or you just train encoder and decoder at the same time with the label of each module?
	4. This is a bit misleading calling it self LLM-based RTL performance prediction. To me, this is just a traditional machine learning based RTL performance prediction with the help from pretrained large language model.

---

> ### Author Rebuttal · Authors · 2025-07-30
>
> > `Q1` The importance of each single line and how much will it affect the backend metrics such as congestion/timing is still beyond my imagination. For example, in Figure. 4, the author provided visualization of actual vs preducted WNS. But what if the backend EDA algorithms changes, will the strong alignment still hold? Such discussion or proof is missing, which will make the statement less convincing. For example, if the placement & routing algorithm changes or even if the technology nodes changes, will the WNS be the same? And for each line in the RTL, the code itself cleary did not change? How will the model adjust to these without additional information? ( And the embedding stage + performance prediction/classification allowing for resource-limited training.)
>
> Thanks for this insightful question. Indeed, as the reviewer points out, while the prediction/classification layers of VeriLoC are specialised for a fixed EDA flow and technology library (details of the flow and tech library in Sec. 4.1), these layers can easily adapt to new flow parameters or tech. library by retraining these layers alone on a training set obtained by running the backend EDA tool (Synopsys RTL Architect in our case) on Verilog code in the training set with these new settings. Only the prediction/classification layers are retrained, which can be done quickly, while the LLM embeddings are generic and reused.
>
> To illustrate this, **we ran a new experiment by rerunning the flow with stricter timing constraints**, resulting in more LoCs with timing and congestion violations. We then retrained the classification/prediction layers on this new data and found that VeriLoC is still able to detect violations with high accuracy. We observe no performance drop with 0.84 F1 for congestion, 0.95 F1 for timing and 0.94 $R^2$ for WNS Prediction.
>
> ---
>
> > `Q2` A following concern is that The model did not provide ablative experiments on the line-level/module-level information. You need to provide it to prove the extra information does help to prediction the features bettern instead of just adopting a stronger model to fit the dataset better.
>
> We appreciate this important observation and draw the reviewer’s attention to Table 5 in the paper that compares “Module-Only” encodings with  “Module + Line-level” encodings. We find that a “VeriLoC-mod” baseline that uses only module-level embeddings performs worse than the full VeriLoC model that uses both line- and module-level features, both in terms of R2 and mean average percentage error (MAPE). In fact VeriLoC-mod is also outperformed by VeriDistill [11], while our full VeriLoC implementation is better than VeriDistill.
>
> We have also **performed new experiments** evaluating a model with line-level encodings only without module encodings. We observe a  drop in performance in congestion F1-score (drop from 0.86 to 0.54) and timing F1-score (drops from 0.95 to 0.68). We will update the paper with these new results. Appendix C.3 also reports results from an alternate architecture where line-level encodings are computed in “batched” fashion.
>
> These results confirm that the performance gains are not due to a stronger model alone, but arise from the combination of local (line-level) and global (module-level) semantic information.
>
> ---
>
> > `Q3` Some more details about the training are not clear to me. How exactly do you label the data of each line-level? How exacly do you train the encoder? or you just train encoder and decoder at the same time with the label of each module?
>
> Thank you for your question. We clarify the two aspects below:
>  - **Line-level labeling:** We use Synopsys RTL Architect (RTL-A), a commercial EDA tool that performs full synthesis and place-and-route and then back-annotates timing slack and congestion metrics at the line level, allowing us to assign precise labels for both classification (e.g., congestion/timing issue or not) and regression (e.g., WNS) tasks.
>  - **Encoder training:** The encoder is trained as part of an autoencoder architecture, as detailed in Appendix B. Specifically, it learns to compress the concatenated line- and module-level embeddings into alower-dimensional latent space and reconstruct them via the decoder. This training is self-supervised and does not use any labels from RTL-A. After training, only the encoder is retained and frozen. Its latent outputs are then used as features for downstream supervised classifiers and regressors.We hope this clarifies both the labeling process and the encoder training procedure.
>
> ---
>
> > `Q4` This is a bit misleading calling it self LLM-based RTL performance prediction. To me, this is just a traditional machine learning based RTL performance prediction with the help from pretrained large language model.
>
> We appreciate the reviewer’s perspective and agree that our method integrates elements of both LLMs and traditional machine learning. To clarify: VeriLoC leverages a pretrained domain-specific large language model from prior work (CL-Verilog [14], a fine-tuned variant of CodeLLaMA) to extract rich, high-dimensional embeddings of Verilog code. These embeddings, derived from the penultimate layer of CL-Verilog, encode both syntactic and semantic information, and are central to our methodology.
>
> That said, we understand the concern and **will revise our wording in the final version** to more precisely characterise the method as based on “*LLM-assisted*” since, as the reviewer notes, we are taking help from a pretrained LLM. While recent work ([11,12])  has also used LLMs in the same way, they only do module level predictions while one of our main contributions is an architecture for line–level predictions. We thank the reviewer for highlighting the need for clearer terminology.
>
> ---
>
> > `Q5` I also like to ask about the training cost, and the overall cost compared with baseline methods.
>
> - Thank you for the question. We summarize the training and inference costs below, along with a clarification regarding available baselines:
>
>   - Embedding Extraction: We use the pretrained CL-Verilog (13B) model to extract embeddings via a forward pass only (no fine-tuning). This process was performed on an NVidia H100 GPU and took under 2 hours for the full dataset.
>
>   - Encoder Training: The encoder for dimensionality reduction (Appendix B) was trained separately in a self-supervised fashion and converged in ~3 hours on an RTX 8000 GPU.
>
>   - Downstream Training: Classifiers like LightGBM and XGBoost were trained on CPU (8 cores, 32GB RAM), requiring only a few minutes due to the reduced feature dimensionality.
>
>   - Inference Cost: Once embeddings are extracted and the encoder is trained, predictions are extremely fast—enabling 14× to 22× average and up to 113× median runtime speedup over full RTL-to-layout flows using commercial tools like Synopsys RTL Architect, as shown in the "Runtime Comparisons" subsection (Section 4.4).
>
>   - Baselines: To the best of our knowledge, no prior work performs line-level prediction of timing or congestion from RTL. Existing methods (e.g., MasterRTL, RTL-Timer, VeriDistill) operate only at the module level, making direct line-level cost or performance comparisons infeasible. This highlights VeriLoC's novelty and the practical importance of its line-level predictions.
>
> - We hope this addresses your concerns regarding training cost and comparative overhead.
>
> ---
>
> > `Q6` I want to know why you chose the exact model architecture?
>
> Thank you for your question. The choice of the proposed model architecture was guided by extensive experimentation with multiple alternatives and careful analysis of their performance. Key insights that motivated the final design are discussed throughout the paper, particularly in Section 4.3 "Timing Prediction and Comparisons with SoTA", Section 4.4 "Discussion", and Appendix C "Additional Ablation Studies". These sections outline the empirical observations and reasoning that informed our architectural decisions.
>
> ---
>
> > `Q7` Do you need to pretrain the encoder or not? Why do you call it LLM when it is just an transformer model? unless it is pretrained on huge amount of general language data？
>
> Thank you for raising this important point. For full clarity, as seen in Figure 2, our architecture consists of three components: (i) a pretrained transformer model (CL-Verilog [14]) from which we extract embeddings, (ii) an encoder for dimensionality reduction, and (iii) a downstream classifier or regressor (e.g., LightGBM).
>
> - The LLM component in our method refers specifically to CL-Verilog [14], a Verilog code generation model from prior work which is obtained by finetuning CodeLLaMA on a large corpus of Verilog code obtained from GitHub. We do not further finetune CL-Verilog on our own data, and use it to extract embeddings of our Verilog code from activations of penultimate layers. As noted in our response to `Q4`, we will use the term “LLM-Assisted” for greater clarity.
>
> - We do train that encoder that performs dimensionality reduction as described in our response to `Q3` and detailed in Appendix B and Section 3.3. The final classification/regression heads are also trained on our curated dataset.

---

> > ### Comment · Reviewer_ryLD · 2025-08-04
> >
> > Thank you for your the detailed rebuttal. The major concerns are addressed. I'll raise my score.

---

### Official Review · Reviewer_BE44 · 2025-07-02

**Clarity:** 4
**Significance:** 3
**Originality:** 4
**Rating:** 5
**Confidence:** 4

**Summary:**

The authors propose using embeddings from Verilog-code generation LLMs to predict routing congestion and timing violations for hardware designs at the line- and module-level. They demonstrate the performance of their method, dubbed VeriLoC, on the OpenABCD RTL/Verilog code dataset using several metrics, including F1 scores for line-level congestions and timing classification, as well as the $R^2$ and mean-average percentage error for the timing violation regression. They compare with several other SOTA methods, and find VeriLoC outperforms them. They also present additional studies related to the impact of Verilog code length and LLM size.

**Questions:**

* How reliable are the line- and module-level labels generated for the dataset? How often are congestion and/or timing violations attributed to the wrong line or module? Would another synthesis tool or different settings give (significantly) different labels?
* I understand the ABCD dataset is public, but will the additional annotations you generated also be made public? This would allow benefit future work.
* How difficult is it to evaluate CircuitFusion on your annotated ABCD dataset? Is the issue that it would require retraining?

**Ethical Concerns:**

["NO or VERY MINOR ethics concerns only"]

**Final Justification:**

My score was already a 5: accept. The authors addressed some concerns I had, and I'm happy with their responses. I'm not sure if the paper merits a score of 6 since the scope is somewhat specific to a particular domain, but I hope my original score is high enough that this paper will be accepted.

**Limitations:**

* Authors should address how much they expect the model to generalize, e.g. to different timing constraints or synthesis tools. Would retraining be necessary for small changes? Consider discussing this in the paper or running some additional studies along this line.

**Quality:**

3

**Strengths And Weaknesses:**

Strengths
* It appears to be the first model addressing the line-level prediction of timing violations and routing congestion, making it quite novel.
* Comparisons to previous SOTA methods, including MasterRTL and RTL-Timer, for the timing violation prediction, indicating improved performance.
* Code is public, improving reproducibility and reusability.

Weaknesses
* No comparison is made to a recent SOTA method called CircuitFusion, where the authors cite difficulty due to the different datasets used for training.
* It's unclear how well this model can generalize to other synthesis tools or timing constraints.

---

> ### Author Rebuttal · Authors · 2025-07-30
>
> > `Q1`  No comparison is made to a recent SOTA method called CircuitFusion, where the authors cite difficulty due to the different datasets used for training. (+ `Q5`: How difficult is it to evaluate CircuitFusion on your annotated ABCD dataset? Is the issue that it would require retraining?)
>
> Thank you for the question! We begin by noting that the main goal/innovation of VeriLoC, line-level prediction, is different from CircuitFusion’s goal of module-level prediction. Therefore, VeriLoC and CircuitFusion cannot be directly compared on this task.
>
> Although not our primary goal, VeriLoC and CircuitFusion can be compared on module-level prediction, but these comparisons are challenging because: (1) CircuitFusion's predictive models use not only RTL code, but also code summaries and post-synthesis netlists which would have to be regenerated on ABCD data; (2) three separate transformer models would have to be finetuned on this data. The GitHub implementation for the Circuit was made available only on April 13th 2025, making it difficult to run comparisons in the limited time available before the NeuRIPS submission deadline (the NeuRIPS deadline for citing past work is March 1st).
>
> For these reasons, we cite CircuitFusion for completeness but refrain from a direct performance comparison. In fact, **we note that VeriLoC and CircuitFusion make orthogonal contributions: both use embeddings of Verilog modules to which VeriLoC concatenates line-level embeddings, while CircuitFusion concatenates embeddings of code summaries and netlists.** As such, two methods could supercharge each other, but we leave that as a topic of future work and will update the paper accordingly.
>
> ---
>
> > `Q2` It's unclear how well this model can generalize to other synthesis tools or timing constraints.
>
> Thanks for this excellent question. VeriLoC can be adapted to new synthesis flows and/or timing constraints by retraining the final classification/regression layers on a training set obtained by re-running the backend EDA tool (Synopsys RTL Architect in our case) with the modified flow settings/constraints. Only the prediction/classification layers are retrained, which can be done quickly, while the LLM embeddings are generic and reused.
>
> To illustrate this, we **ran a new experiment with a stricter timing constraint of 0.2 ns** as the reviewer requested, resulting in more LoCs with timing and congestion violations. We then retrained the classification/prediction layers on this new data and found that VeriLoC is still able to detect violations with high accuracy. We observe qualitatively similar performance with 0.84 F1-score for congestion, 0.95 F1-score for timing, and 0.94 $R^2$ for WNS Prediction.
>
> ---
>
> > `Q3` How reliable are the line- and module-level labels generated for the dataset? How often are congestion and/or timing violations attributed to the wrong line or module? Would another synthesis tool or different settings give (significantly) different labels?
>
> The ground-truth labels generated are from a commercial EDA tool (Synopsys RTL Architect) that actually performs synthesis and physical design, and then steps backwards from the final result to arrive at line-level annotations. While incorrect annotations cannot be ruled out at such an early stage, we did not observe this in our data. We note that different tool settings do give different results, as noted in our prior response (Q2) regarding stricter timing constraints.
>
> ---
>
> > `Q4` I understand the ABCD dataset is public, but will the additional annotations you generated also be made public? This would allow benefit future work.
>
> - Yes, all the source code and dataset generated for this work will be open-sourced.

---

> > ### Comment · Reviewer_BE44 · 2025-08-07
> > **Addressed comments**
> >
> > Thank you for the clarifications as they have fully addressed my comments, especially on how well the tool generalizes to other tools or timing constraints. I would ask that the paper be updated with this additional study and the same clairifications on only needing to retrain the final layers (or at least added to an appendix).

---

### Decision · Program_Chairs · 2025-09-17

**Decision:**

Accept (poster)

**Comment:**

This paper introduces VeriLoC, the first framework to predict hardware design quality at both the line-level and module-level directly from Verilog, addressing both congestion and timing. The paper makes a clear and original contribution to early-stage hardware design quality prediction, extending LLM-based techniques to the novel and practically relevant line-level setting. In the rebuttal discussion, the author made clear clarifications, added new experiments with stricter timing constraints, ablation studies, and training/inference cost. The reviewers agreed that their major concerns were resolved and are all in favor of acceptance. I recommend acceptance, and hope the authors could incorporate the additional experimental results and clarifications into the camera-ready version.